# Molecular Phylogeny of Cimicoidea (Heteroptera: Cimicomorpha) Revisited: Increased Taxon Sampling Reveals Evolution of Traumatic Insemination and Paragenitalia

**DOI:** 10.3390/insects14030267

**Published:** 2023-03-08

**Authors:** Sunghoon Jung, Junggon Kim, Ondřej Balvín, Kazutaka Yamada

**Affiliations:** 1Laboratory of Systematic Entomology, Department of Applied Biology, College of Agriculture and Life Sciences, Chungnam National University, Daejeon 34134, Republic of Korea; 2Department of Smart Agriculture Systems, College of Agriculture and Life Sciences, Chungnam National University, Daejeon 34134, Republic of Korea; 3Department of Ecology, Faculty of Environmental Sciences, Czech University of Life Sciences Prague, Kamýcká 129, 165 21 Prague, Czech Republic; 4Institute of Natural and Environmental Sciences, University of Hyogo/Museum of Nature and Human Activities, Yayoigaoka 6-chome, Sanda-shi 669-1546, Hyogo, Japan

**Keywords:** Cimicoidea, evolution, Heteroptera, paragenitalia, phylogeny, traumatic insemination

## Abstract

**Simple Summary:**

The superfamily Cimicoidea comprises seven families with more than 600 described species. The members of this group show two feeding habits, predation and blood-feeding, and show the unique hypodermic insemination process called traumatic insemination. A molecular phylogenetic study using increased sampling aimed to hypothesize the phylogenetic relationships within Cimicoidea, and to understand the evolutionary history of traumatic insemination and the correlation between the insemination habit and the morphology, paragenitalia. The phylogenetic results showed that most families within Cimicoidea were confirmed as monophyletic groups and hypothesized the novel sister-group relationship of Curaliidae + Lasiochilidae with high support values. Additionally, it was revealed that at least one shift from standard insemination to traumatic insemination occurred within Cimicoidea, and the acquisition of paragenitalia in cimicoid females was correlated with the traumatic insemination habit.

**Abstract:**

The molecular phylogeny of the Cimicoidea was reconstructed from an expanded sampling based on mitochondrial (16S, COI) and nuclear (18S, 28SD3) genes. The data were analyzed using maximum likelihood (ML), maximum parsimony (MP), and Bayesian inference (BI) phylogenetic frameworks. The phylogenetic relationships inferred by the model-based analyses (ML and BI) were largely congruent with those inferred by the MP analysis in terms of the monophyly of most of the higher taxonomic groups and the species-level relationships. The following clades were recovered in all analyses: Cimiciformes; Nabidae: Prostemmatinae; Nabidae: Nabinae; Plokiophilidae; Microphysidae; Lasiochilidae; Cimicidae: Cacodminae; Cimicidae; Lyctocoridae; *Anthocoridae s. str*.; Cardiastethini excluding *Amphiareus*; Almeidini; Scolopini; Anthocorini; Oriini; Curaliidae + Lasiochilidae; Almeidini + Xylocorini; Oriini + Cardiastethini; and Anthocorini + *Amphiareus*. Reconstructions of ancestral copulation states based on Bayesian and parsimony inference indicated that at least one shift from standard insemination (SI) to traumatic insemination (TI) occurred within Cimicoidea, and an investigation of the evolutionary correlation between TI and paragenitalia (PG) revealed that the acquisition of PG in cimicoid females was correlated with the TI habit. Additionally, our morphological examination of various types of PG suggested that even the same PG type may not constitute a homologous feature at various taxonomic levels, indicating the convergent evolution of female morphology to adapt to TI.

## 1. Introduction

The superfamily Cimicoidea encompasses seven families with more than 600 described species [1,2,3]. Members of this group show two feeding habits, predation and blood-feeding [1], and live in diverse microhabitats [3,4,5]. This group includes two medically and economically important families, the bedbugs or bat bugs (medical pests; Cimicidae) and the flower bugs or minute pirate bugs (biological control agents; Anthocoridae) (Figure 1a,b, bedbugs; Figure 1c–e, flower bugs).

### 1.1. Phylogenetic Relationships

Ford (1979) [6] conducted the first phylogenetic study of Cimicoidea using 14 morphological and biological characters. This analysis found Cimicoidea to be monophyletic and Lasiochilidae and Plokiophilidae to be basal groups within Cimicoidea. More recently, two major phylogenetic studies including members of Cimicoidea have been conducted. (i) Schuh et al. (2009) [2] proposed higher-level relationships within the Cimicomorpha (Hemiptera: Heteroptera) based on phylogenies constructed using morphological and molecular data. Their study included a relatively large sample of Cimicoidea and indicated that cimicoids were monophyletic; however, *Anthocoridae s. str.* were paraphyletic due to the position of Lyctocoridae. (ii) Jung et al. (2010) [3] proposed higher-level relationships within *Anthocoridae s. lato*, including a significant sampling of Cimicoidea, based on molecular data (16S, 18S and 28S rRNA). They also found the cimicoids to be monophyletic and to be the sister group to Nabidae. In addition, they presented new higher-level phylogenetic relationships within *Anthocoridae s. lato*, such as Oriini + Cardiastethini and Cimicidae + Plokiophilidae. Both studies agreed that Lasiochilidae was the most basal group within Cimicoidea; however, most of the proposed higher-level phylogenetic relationships within Cimicoidea remain controversial and require further study (e.g., the sister-group relationships of Cimicidae + Plokiophilidae and Cimicidae (including Polyctenidae) + Curaliidae; the positions of Cimicidae and Lyctocoridae; etc.) [2,3,7]. Additionally, both studies were lacking in terms of taxon sampling because certain higher taxonomic groups were not included or were represented by only one species (e.g., Plokiophilidae).

### 1.2. Traumatic Insemination and Paragenitalia

Traumatic insemination (TI), also known as hypodermic insemination, is the process of insemination through the body wall into the body cavity (haemocoel) rather than into the female’s genital tract, resulting in the physical breaching of the epidermis [1,6,8,9,10,11] (Figure 2A–C). In Cimicoidea, the sperm spreads through the female’s hemolymph, reaches the ovaries or vitellarium [6], and effects fertilization [1,10]. This unique behavior has long been studied and has attracted evolutionary biologists hoping to understand co-evolutionary “arms races” with special reference to sexual conflict between the sexes [10]. In addition to male–female couplings, TI has occasionally been observed in cases of homosexual and interspecific insemination [8,10,12,13]. Although TI seems to have evolved independently in many invertebrate groups and species [10,14], it is not a common behavior within the animal kingdom. Among the true bugs (Hemiptera: Heteroptera), TI is most prevalent in the superfamily Cimicoidea [15] and in the mirid genus *Coridromius* [16]. Additionally, most species belonging to the subfamily Prostemmatinae (Nabidae) are known to engage in TI [15]. In Cimicoidea, TI has been most thoroughly studied in the common bedbug, *Cimex lectularius* [10,11,15], which is a notorious, medically significant pest of humans. In some bedbugs, damage to the female from TI can affect longevity and reproductive success [11].

The females of many cimicoid species that practice TI have developed specialized morphological organs called “paragenitalia” (PG) [9,15,21]. Carayon (1977) [9] subdivided PG into three morphological types: (i) ectospermalege, (ii) copulatory tube, and (iii) omphalus (Figure 2D–G). Among these structures, the spermalege is known to reduce physical damage, infection risk, and other potential side effects of TI in certain bedbugs [10,22]. Carayon (1959; 1966) [15,21] asserted that these organs must be related to TI; however, the exact functions of each PG type have received little study in most cimicoids, except for some bedbugs (e.g., [11]). Recently, Horton and Lewis (2011) [23] reported that the copulatory tube (Figure 2E) in some *Anthocoris* species (Anthocoridae) plays a role in receiving male genitalia and in storing the sperm temporarily. However, in contrast to Carayon’s predictions [15], several taxonomic studies have reported that some cimicoid species and higher taxonomic groups that engage in TI do not possess any PG [4,6,19,24,25].

Because many biological and systematic questions in Cimicoidea remain to be addressed, the main goal of this study is to use more molecular data and denser taxon sampling (compared to previous studies) to reconstruct the molecular phylogeny of the Cimicoidea. Using this phylogeny, we infer the ancestral character states of TI and test the correlations between TI and PG. Based on our results, we present the evolutionary history of TI in Cimicoidea with special reference to PG.

## 2. Materials and Methods

### 2.1. Taxon Sampling

Because of the small size and similar external shape of cimicoids, we dissected the majority of the specimens used in this study prior to molecular work to confirm the genital characteristics, which are the most important characteristics for species-level identification [24,25,26,27,28].

In total, 53 taxa comprising 41 ingroup terminals and 12 outgroup taxa were included in this study (Table 1). Ingroup sampling included six of the seven existing cimicoid families: Lasiochilidae, Anthocoridae, Lyctocoridae, Plokiophilidae, Cimicidae, and Curaliidae. Outgroup sampling included representatives of the families within Cimiciformes: Microphysidae, Joppeicidae, and Nabidae (Nabinae and Prostemminae) as sister species to Cimicoidea [2]. Reduviidae within the infraorder was used to root the tree based on currently accepted cimicomorphan relationships [2]. In particular, we included new molecular data for several cimicoids that have not been obtained previously (e.g., plokiophilids and cimicids; 20 sequences in 18S rRNA, in 28S rRNA, and in 16S rRNA; 21 sequences in COI; Table 1). Additionally, we sampled all tribes in the controversial family *Anthocoridae s. lato* in order to address questions (see Jung et al. (2010) [3] for details).

### 2.2. Molecular Markers, DNA Extraction, and Other Molecular Protocols

The molecular data consisted of two nuclear ribosomal (complete 18S rRNA and partial 28 rRNA-D3 region), one mitochondrial ribosomal (16S rRNA), and one mitochondrial protein-encoding (COI) markers. These markers were informative in earlier studies of phylogenetic relationships within cimicoids and mirids (Cimicomorpha), as well as within higher-level cimicomorphan families [2,3,30,31].

The genomic DNA was extracted from single individual specimens using GENEALL (Exgene Tissue SV; Geneall, Seoul, Republic of Korea) in accordance with the manufacturer’s protocol. After boring a hole in the exoskeleton, the sample was put in AE buffer with proteinase K for approximately 24 h. After incubation, the exoskeleton sample and a genital segment from all specimens were made as the voucher macerated slide specimens [24,29]. Specimens were lodged in the laboratory of Systematic Entomology in Chungnam National University (CNU).

Polymerase chain reactions (PCRs) were conducted using Advantage PCR II Taq polymerase (BD Advantage™, San Jose, CA, USA) according to the general protocol: in a 20 μl reaction mixture containing 0.4 μm of each primer, 200 μm dNTPs, 2.5 μm MgCl_2_, and 0.05 μg genomic DNA template. The thermal cycling process consisted of 40 cycles of 92 °C for 30 s, 43–52 °C for 30 s, and 72 °C for 60 s, followed by a final extension at 68 °C for 10 min. The PCR products were purified using a QIAquick^®^ PCR purification kit (QIAGEN, Hilden, Germany), and directly sequenced at NICEM (National Instrumentation Center for Environment Management, Seoul National University, Republic of Korea). The information of the primer set is shown in Table 2. New sequences have been deposited in GenBank (Table 1).

### 2.3. Alignments and Phylogenetic Analyses

The molecular sequence data set used for the analysis comprised a total of 3935 bp: 2024 bp of 18S rRNA, 755 bp of 28S rRNA, 498 bp of 16S rRNA, and 658 bp of COI. The sequences were arranged using Sequencher 4.0 and 4.9 (Gene Codes Corp., Ann Arbor, MI, USA) and have been deposited in GenBank (Table 1). Nucleotide sequences were manually aligned using Se-Al, version 2.0a11 [37]. Additional alignments for the rRNA genes were performed using MAFFT [38,39] separately by gene partitions (16S, 28S, and 18S) via the online server (v.6; http://align.bmr.kyushu-u.ac.jp/mafft/online; accessed on 13 February 2018). The Q-INS-I strategy was selected for rRNA genes, which considered RNA secondary structure and small data sets (<200) [40], with ambiguous regions as identified by Gblocks ver. 0.91b [41] eliminated. The following relaxed parameters were used in Gblocks for each gene partition of the rRNA genes: minimum number of sequences for a conserved position = 72 (n × 0.5), minimum number of sequences for a flanking position = 72 (n × 0.5), maximum number of contiguous non-conserved positions = 50, minimum length of a block = 5, and allowed gap positions = “with half”.

As COI sequences had no indels, the protein-coding genes were translated to amino acids in MacClade 4.05 [42] for examination and refinement of the nucleotide alignment and were additionally aligned using the FFT-NS-I strategy implemented by the MAFFT online server. Prior to running phylogenetic analyses, we assessed partition homogeneity using the incongruence length difference (ILD) test [43] for the combined data set of four gene regions (18S, 16S, 28S and COI) implemented in PAUP ver. 4b10 [44].

Parsimony analyses were conducted with TNT ver. 1.1 for Windows [45]. New technology searches [46,47] were used, the memory was set to hold 10,000 trees, and the default settings for sectorial search, drift, ratchet, and tree fusing were used, as well as ten initial additional sequences and the option to find minimum length trees ten times. In all parsimony analyses, gaps were treated as missing data, consistent with the standard gap treatment in the likelihood and Bayesian analyses. One thousand replications for jackknife resampling (removal probability set at 36) were performed.

Bayesian inference (BI) was also used to estimate phylogenetic relationships. The models for nucleotide substitution used in the analyses were selected for each marker individually by applying the Bayesian Information Criterion as implemented by MODELTEST ver. 3.7 [48] in conjunction with PAUP* ver. 4.0b10 [44]. The evolutionary model selected for each molecular marker was a general time-reversible plus invariant sites plus gamma-distributed model (GTR + I + G). Three heated chains and one ‘‘cold’’ chain were used. Posterior probabilities of trees and parameters in the substitution models were approximated with a Markov chain Monte Carlo method using MRBAYES ver. 3.1.2 [49]. Search settings used the default parameters. Like TNT analyses, the combined analysis included all four molecular markers (evolutionary model: GTR + I + G). In the combined analysis, each partition was modeled independently (unlinked models). All chains were run for 10 million generations, with trees sampled every 100 generations from the cold chain. The sampling frequency was every 100th generation and 25% of the samples were summarized. The results of the BI inference analyses with posterior probabilities (PP) are presented as 50% majority-rule consensus trees.

Lastly, the molecular data set was also analyzed using maximum likelihood (ML) through the CIPRES Portal 1.15 (http://www.phylo.org/sub_sections/portal/; accessed on 13 February 2018) using RAxML bootstrapping [50]. In the analysis, each partition was modeled independently (unlinked models; evolutionary model: GTR). Default settings for a maximum likelihood search with 1000 bootstrapping runs were used with a random seed for bootstrapping of 12345 and a random seed for parsimony inferences of 23456. The Reduviidae taxon was used to root the tree for all analyses.

### 2.4. Reconstructing Ancestral Character States

#### 2.4.1. Copulating Types

Among ingroups, the family Lasiochilidae is the only known family within the Cimicoidea to copulate normally (standard insemination; SI) [1,6,15]. All cimicoid species except Lasiochilidae have been reported to be engaged in TI when copulating [9]. However, the copulating behavior of the monotypic and recently discovered family Curaliidae is still unknown as the female of the family has not been discovered yet [51]. Although Schuh et al. (2008) [51] assumed that this species would copulate in a normal way based on their observation of the male genital morphology, we coded this species as “uncertain” in terms of the copulation behavior. Additionally, Schuh (2006) [52] has recently argued that *Heissophila macrotheleae* (Plokiophilidae) would copulate in a normal way due to a lack of TI-related morphological characteristics (e.g., sickle-like male paramere (Figure 2C), copulatory tubes (Figure 2D,E), and scars after TI), which is contrary to the hypothesis proposed by Carayon [9,15] that all plokiophilids are engaged in TI. Therefore, the copulating habit of the Plokiophilidae is controversial, and possibly varies at the species level, thus the taxa of the Plokiophilidae used in this study were coded as “uncertain” in terms of their copulating behavior (Table 1). Please note that the plokiophilid females in this study do not have any PG and those are possibly new species (direct observation by the first author).

Among outgroup taxa, most prostemmines (Nabidae) inject spermatozoids into the haemocoel by insertion of the male genitalia into the female genital opening followed by penetration of the vaginal wall [6,8]. Therefore, we coded those taxa used in this study as TI (Table 1), as the copulation type of this nabine group is not a standard type, although it may be considered different to the other TI types.

#### 2.4.2. Reconstructions of Ancestral Character States

A Bayesian approach, as implemented in the BayesTraits 2.0. [53] software package, was used to reconstruct ancestral character states of TI for selected nodes in the molecular phylogeny (BI 50% majority-rule consensus tree). BayesTraits uses reversible-jump MCMC methods to derive posterior probabilities and the values of traits at ancestral nodes of the phylogeny [54]. BayesMultiState was selected as the model of evolution and MCMC as the method of analysis. The rate deviation was set to 10. A hyperprior approach was employed with an exponential prior seeded from a uniform prior in the interval 0–10. Thus, acceptance rates in the preferred range of 20–40% were achieved as recommended [55]. A total of 50 million iterations were run for each analysis with the first 1 million samples discarded as burn-in, with sampling every 1000th generation. Because the posterior probabilities for ancestral patterns of the single runs differed slightly, we calculated the arithmetic mean of all samples for reconstruction of ancestral types. The posterior probabilities for ancestral patterns were mapped on the ML tree as pie charts on the nodes.

We used the result from the parsimony analysis as a reference topology to obtain the ancestral parsimonious character states. The reconstructions were optimized using the parsimony ancestral states (PAS) option in Mesquite ver. 2.6 [56]. The type codes of the feeding habit used in the BayesMultiState analysis were applied to the parsimony character states analysis.

### 2.5. Testing Correlations

#### 2.5.1. Paragenitalia and Copulation Types

Paragenitalia (PG) can be subdivided into three types morphologically by Carayon (1977) [9]: (i) ectospermalege (coupled with mesospermalege in some cimicoid species, mostly in the Cimicidae and some in the Xylocorini [4,57]; Figure 2A,F); (ii) copulatory tube or double copulatory tubes (in several tribes, such as the Scolopini, Anthocorini, and Oriini in the Anthocoridae [57]; Figure 2D,E, respectively); and (iii) omphalus (most species in the tribe Cardiastethini, “Omphalophore Dufouriellini” [6]; Figure 2G). Therefore, we regarded a species as having PG when the species contained one of these three morphological characters (Table 1). Please note that we double-checked most of the specimens in this study directly by our observation after dissection in addition to taxonomical references. For copulating types see Section 2.4.1.

#### 2.5.2. Correlation Tests

We tested associations between characters using a maximum likelihood model to analyze correlations between discrete characters across evolutionary trees [54]. To evaluate the correlation, BayesTraits 2.0. [53] was also used. All sampled 8000 trees from the above BI analyses were used as the input tree file for the program and the correlation test [58]. An additional two-character data set (TI and PG) for the terminal taxa was prepared as the input file. This analysis was implemented in the Discrete module of BayesTraits. The log likelihood score of a model of evolution in which two traits evolve independently across the phylogeny was compared with the log likelihood score of the model enforcing correlated evolution of the traits [54]. The *p* value for the comparison of these two models of evolution can be obtained by a likelihood ratio test: 2[(log likelihood (dependent model) − log likelihood (independent model))] (2dLnlike), which has been shown to approximate a chi-squared distribution with four degrees of freedom [54,55]. Furthermore, to test the correlation, independent and dependent models can be compared using the harmonic mean, calculated as 2[log (harmonic mean (dependent model)) − log (harmonic mean (independent model))] (=log–Bayes factor), as implemented in BayesTraits. The log-Bayes factor (BF) derived from the harmonic means of the likelihoods can be interpreted as follows: >2, some evidence that the dependent model is favoured; >5, strong evidence that the dependent model is favoured; >10, very strong evidence that the dependent model is favoured [55]. For the analysis, several MCMC runs were performed for both models using 2,000,000 iterations. The MCMC chain was sampled every 100 iterations after a burn-in of 1000 iterations. To achieve an acceptance rate of between 20 and 40%, a ratedev value of 2 or 8 was chosen individually for the independent or dependent model. A hyperprior procedure was used to establish the parameters for an exponential prior. A reversible jump was applied, seeding the exponential prior from a uniform prior in the interval 0–30.

## 3. Results

### 3.1. Phylogenetic Relationships

Topologies based on analyses of sequences from each partition exhibited no significant conflict (ILD tests for each data combination of 18S, 28S, 16S, and COI, *p* > 0.05). The TNT analysis produced one most parsimonious tree (*L* = 6348; Ci = 0.38; Ri = 0.64), which is shown in Appendix A with jackknife supporting values. The final ML optimization likelihood score was –27,906.513772 (ML tree is shown in Figure 3). The Bayesian analysis tree (a 50% majority-rule consensus tree) contains the branches almost identical to ML topologies except for specific clades with collapsed branches (=polytomies), which is shown in Appendix A. Jackknife supporting values, ML bootstrapping values, and PP were generally strong for most monophyletic groups, but were relatively weak or moderate for higher-level phylogenetic relationships (Figure 3; Appendix A). The parsimony topology was different from the topologies estimated from the ML and the BI analyses with regard to the positions of Plokiophilidae, Blaptostethini, and Lyctocoridae among ingroups. Thus, monophyly of TI-cimicoids was supported in ML and the BI, but not in the parsimony analysis as Blaptostethini sistered to Curaliidae + Lasiochilidae (Appendix A; Figure 3 in red color). In the parsimony analysis, the family Plokiophilidae was positioned out of the Cimicoidea clade, and the family Nabidae was divided into two groups as a non-monophyletic group: two subfamilies, Prostemmatinae and Nabinae (Appendix A).

Several clades were supported in all analyses, while some clades were recovered in only a subset of analyses. The unambiguously supported clades were characterized with high support values (Figure 3; Appendix A). The higher taxon clades were supported in all analyses and were mostly found to be monophyletic as follows: Cimiciformes; Nabidae: Prostemmatinae; Plokiophilidae; Microphysidae; Nabidae: Nabinae; Lasiochilidae; Cimicidae: Cacodminae; Cimicidae; Lyctocoridae; Scolopini; Anthocorini; Oriini; and Xylocorini (Figure 3; Appendix A). Several phylogenetic relationships of higher taxonomic groups were also supported in all analyses: (i) the tribe Cardiastethini was separated into two distinct groups, the genus *Amphiares* and the rest of Cardiastethini, which are clustered with the tribes Anthocorini and Oriini, respectively; (ii) a sister-group relationship of the two families, Curaliidae + Lasiochilidae; (iii) a sister-group relationship of the two tribes, Xylocorini + Ameidini (Figure 3; Appendix A).

### 3.2. Reconstructions of Ancestral TI States

The ancestral copulating types were estimated for the six nodes of higher taxa relationships, which were mapped on the ML tree (Figure 3; pie charts). The analysis suggested that SI at the root of Cimiciformes had a reconstructed probability of greater than 99% compared to the alternative type (Figure 3, the node of the Cimiciformes). TI was reconstructed on the node A, the Cimicoidea excluding the two families, Curaliidae and Lasiochilidae, with a probability of greater than 99% (Figure 3). The origin of copulating type for the common ancestor of the Cimicoidea is not so clear; it was reconstructed as TI with a probability of about 70% (Figure 3; indicated by Cimicoidea with an arrow). The parsimony reconstructions optimized by using the PAS option are shown in Appendix A, which were largely congruent with the observed Bayesian character reconstructions (Figure 3); both of the analyses indicated that TI evolved only once within the Cimicoidea.

### 3.3. Bivariate Evolutionary Correlations

A likelihood ratio test confirmed the correlation between PG and TI (2dLnlike = 11.29, *p* = 0.023; independent model = −26.234, dependent model = −20.587). The BayesFactor tests also confirmed the evolutionary correlation between PG and TI (*BF* = 7.02; RJ-MCMC harmonic mean of independent = −32.248; RJ-MCMC harmonic mean of dependent = −28.734). Because harmonic means may be unstable, we repeated the analysis several times. In all runs, the magnitude of the difference between the harmonic means was very similar; these repeats consistently support the correlation between PG and TI. The results of the two correlation tests confirmed the evolutionary correlation between PG and TI.

## 4. Discussion

### 4.1. Phylogenetic Incongruence among Analyses

The phylogenetic result of the parsimony analysis (Appendix A) was largely congruent with those of the model-based phylogenetic analyses (ML and BI) in terms of the recovered monophyly of major families and tribes and the species-level relationships within each higher taxonomic group. However, the supporting nodes for relationships among higher taxa within Cimicoidea (e.g., at the family level) were relatively poor in the parsimony analysis (Appendix A) and showed topological incongruence compared to the ML and BI trees (Figure 3; Appendix A). The model-based analyses (BI and ML) yielded nearly identical results, except that the BI reconstructed polytomies with relatively high support values (Figure 3; Appendix A) at certain nodes representing relationships among higher taxa, which may result from the models applied to each analysis. There were two major areas of incongruence between the MP and model-based analyses: (i) Plokiophilidae was located outside the main Cimicoidea clade in the MP tree, indicating that Cimicoidea might not be a monophyletic group. In the ML and BI trees (Figure 3), however, Plokiophilidae was sister to the remaining cimicoids (sister to the Lasiochilidae + Curaliidae clade), supporting the current circumscription of Cimicoidea [2]. This incongruence between MP and model-based analyses, and the positions of the Plokiophilidae and the cimicoid families were also reconstructed in a recent phylogenetic study [59]. (ii) The tribe Blaptostethini (Anthocoridae) was clustered with the Lasiochilidae + Curaliidae clade in the MP tree, and its placement also received low support values in the model-based analyses (Figure 3; Appendix A). We found several sister-group relationships among higher-level taxa, such as Cimicidae + (Blaptostethini + Curaliidae + Lasiochilidae) and Lyctocoridae + (Almeidini + Xylocorini + Scolopini) in the MP tree (SI1) and Lyctocoridae + Cimicidae + (Scolopini and Blaptostethini) in the BI and ML trees (Figure 3; Appendix A). However, all analyses yielded low support values for nodes representing the phylogenetic relationships of higher taxonomic groups, indicating the need for further research using additional morphological and molecular data (e.g., nuclear protein-coding genes) and more robust sampling.

### 4.2. Phylogenetic Relationships within Cimicoidea

Despite the incongruence between the MP and model-based analyses, four major phylogenetic results were supported by all analyses (Figure 3; Appendix A). First, most families within Cimicoidea were confirmed as monophyletic groups, including Lasiochilidae, Cimicidae, Lyctocoridae, and Plokiophilidae, as were most tribes within *Anthocoridae s. lato* (e.g., Xylocorini, Almeidini, and Scolopini). Second, the novel sister-group relationship of Curaliidae + Lasiochilidae received high support values (Figure 3; Appendix A). According to Schuh et al. (2008) [51], these two groups copulate in a normal manner, unlike other cimicoids [9,21,51]. Thus, the monophyly of standard insemination is also supported in Cimicoidea (Figure 3), although we codified Curaliidae as uncertain for the copulation type. Note that Schuh et al. (2008) [51] hypothesized that Curaliidae engage in standard insemination based on the morphological characters of the male genitalia. Third, the monophyly of *Anthocoridae sensu* Jung et al. (2010) [3] also received relatively high support values in all analyses (PP = 1.00/jackknife = 70/ML bootstrap = 98; Figure 3, node B; Appendix A). Fourth, *Anthocoridae s. lato* (also *sensu* Carayon (1972)) [57] was not monophyletic; thus, our results do not support the classification systems of Carayon (1972) [57] and Štys and Kerzhner (1975) [60]. Although the classification systems and taxonomic ranks within this family have long been controversial (*Anthocoridae sensu lato*; see Jung et al. (2010) [3] for a detailed historical review), our model-based molecular phylogeny largely supported the classification system of Schuh and Štys (1991) [7] (e.g., with regard to the position of Lasiochilidae and Lyctocoridae) among the various hypotheses for *Anthocoridae sensu lato* (e.g., [57,60,61,62]) (Figure 3, Appendix A). Within the family *Anthocoridae s. lato*, the phylogenetic positions of the tribes Blaptostethini and Scolopini remain ambiguous [3] because all analyses placed these taxa outside the main clade of *Anthocoridae s. lato* with low support values (Figure 3, Appendix A). Additionally, all analyses supported the sister-group relationship of Xylocorini + Almeidini, which was proposed by Ford (1979) [6] based on the apomorphic character of seven testicular lobes (Figure 3, Appendix A). However, our results rejected the sister-group relationship of Anthocorini + Oriini, also proposed by Ford (1979) [6], as the only group of facultative pollen feeders within Cimicoidea. These taxa were clustered with the genus *Amphiareus* and the tribe Cardiastethini, respectively, in all analyses (Figure 3, Appendix A).

Compared to other recent phylogenetic studies, our results do not support the hypothesized sister-group relationships of Xylocorini + Cimicidae + Plokiophilidae [3], Lyctocoridae + Oriini [2], and Lyctocoridae + (Plokiophilidae + Oriini) + (Cimicidae + Anthocorini) [59] because Oriini was clustered with Cardiastethini (except for *Amphiareus*) in all analyses (Figure 3; Appendix A). Additionally, all analyses did not support that Lasiochilidae is a sister group to all remaining cimicoids (Figure 3, Appendix A), in disagreement with the findings of previous phylogenetic studies [2,3,6,7]. The sister familial group of the Lasiochilidae within Cimicoidea was unknown in Jung et al. (2010) [3], while the Lasiochilidae was found to be closely related to the family Curaliidae in all analyses (MP, BI, and ML) in this study. The model-based analyses indicated that Plokiophilidae is the sister group to all remaining groups within Cimicoidea (Figure 3). This is congruent with the results from model-based analyses in Kim et al. (2022) [59] but is incongruent with the results from the preferred topology within Cimicomorpha in Schuh et al. (2009) [2], and the results from the previous study in Jung et al. (2010) [3], respectively.

### 4.3. Evolution of Traumatic Insemination with Special Reference to Paragenitalia

The most common ancestral state of traumatic insemination (TI) was reconstructed at node A within Cimicoidea with more than 99% posterior probability (Figure 3, pie chart; Appendix A). Thus, this behavior evolved once from the ancestral state of standard insemination (Figure 3, Cimiciformes node), also indicating that the TI group (*Anthocoridae s. lato* + Cimicidae + Lyctocoridae) within Cimicoidea is monophyletic (Figure 3, node A). Within Cimiciformes, TI evolved independently from SI, the ancestral copulating habit of Cimiciformes, at node C, Nabidae: Prostemmatinae. Thus, TI evolved at least twice within Cimiciformes (Figure 3; Appendix A). Meanwhile, the family Plokiophilidae currently comprises two subfamilies (Plokiophilinae and Heissophilinae), and the state of lacking traumatic insemination is one of the diagnostic characters for Heissophilinae [63], although the copulating habit possibly varies at the species level. This family was recovered as monophyletic and was closely related to the clade Cimicidae + (Polyctenidae + Curaliidae) [2]. However, the position of this family within Cimicoidea is not strong, considering the conflict of the placement from the previous studies (e.g., [2,59]). This suggests that the process of the evolution of TI from SI within Cimicoidea should be reconsidered after the monophyly and the placement of Plokiophilidae are reevaluated and/or the ancestral state reconstruction of insemination is tested within Plokiophilidae. In this way, the multiple origin of TI within Cimicoidea or Cimiciformes will be re-hypothesized based on the result of further research.

Our results also confirmed an evolutionary correlation between TI and PG. The TI habit is associated with the development of PG in this insect group (Figure 3; taxa in boldface). In addition to the results of the correlation tests, all cimicoids with PG engage in TI (Figure 3; red-colored taxa in boldface), indicating that non-TI cimicoids have never developed PG. Therefore, we reinforce that the acquisition of various PG types by female cimicoids represents a defensive mechanism against TI, supporting Carayon’s (1977) [9] hypothesis. Such a role has been demonstrated for the spermalege of common bedbugs [10]. Recently, Horton and Lewis (2011) [23] showed that the copulatory tube of some *Anthocoris* species in North America receives the male intromittent organ (endosoma). Additionally, among true bugs, a recent study confirmed the sexual co-evolution of TI behavior and PG complexity in the genus *Coridromius* (Heteroptera: Miridae) [64].

Although each higher taxonomic group of cimicoids that exhibits TI behavior appears to have evolved its own PG type (e.g., the copulatory tube in the genera *Orius* and *Anthocoris* and the spermalege in the family Cimicidae and the tribe Xylocorini [9]) (Figure 2), some species within groups and some entire higher taxonomic groups (e.g., the genus *Amphiareus*) lack PG. For example, *Xylocoris flavipes* lacks PG, while *X. cerealis* has ectospermalege-type PG similar to those found in cimicids [9,19]. Additionally, within Cimicidae, *Aphrania elongata* lacks PG [4], and within the tribe Cardiastethini, *Physopleurella armata* lacks PG [26], unlike most other species within the tribe [9,15] (Figure 3; Table 1). Therefore, each higher taxonomic group within Cimicoidea has evolved its own type of PG, and some groups have not developed any PG (e.g., the family Lyctocoridae, the tribe Almeidini, and the genus *Amphiareus*; [9,15,24]. Additionally, the possession of PG varies at the species level in some groups within Cimicoidea.

As shown in Figure 2, the three types of PG are not homologous in position and structure at all taxonomic levels (from species to family). Moreover, even the same type of PG may not be homologous in some cases. For example, the copulatory tubes of Anthocorini are usually elongated and are associated with a relatively large sperm storage organ (Figure 2D); however, those of Oriini are shortened and are not associated with a sperm storage organ (Figure 2E) [27,28]. Additionally, the openings of the copulatory tubes in some plokiophilids (e.g., *Embiophila africana*) [9] are located on the dorsal surface of the second to third abdominal segments on each side, whereas those in Anthocoridae are located internally on the medioventral side of the sixth to seventh abdominal segments. The two undetermined terminal taxa within Plokiophilidae that were sampled in this study (possibly new species) have no copulatory tube (direct observation). Therefore, the position and structure of the copulatory tube vary at each taxonomic level, even at the species level. Similarly, the spermaleges of Cimicidae are variable. Although Carayon (1966) [15] attempted to categorize the position and shape of the spermalege at the genus level within Cimicidae (Figure 2A,F), these features sometimes appear random at the genus and even species level (e.g., among species in the genus *Leptocimex*) [4]. Additionally, the evolution of PG within Xylocorini is unique. Not only do some species within this higher taxon lack PG, but other species possess different types of spermalege (e.g., *Xylocoris flavipes*) [9]. For example, *X. cerealis* possesses ectospermalege-type PG similar to those found in cimicids, while *X. galactinus* has copulatory tubes [9,19]. The position of the omphalus also differs at the species level (Figure 2G). For example, the omphalus of *Cardiastethus exiguus* is located on the anterior part of the seventh sternum, whereas that of *Buchanaiella anulata* is located on the medial part of the seventh sternum [9]. Therefore, the three types of PG are homoplastic characters within Cimicoidea (Figure 2), and even the same PG type should not be assumed to represent a homologous structure but rather to result from convergent morphological evolution in response to TI.

## Figures and Tables

**Figure 1 insects-14-00267-f001:**
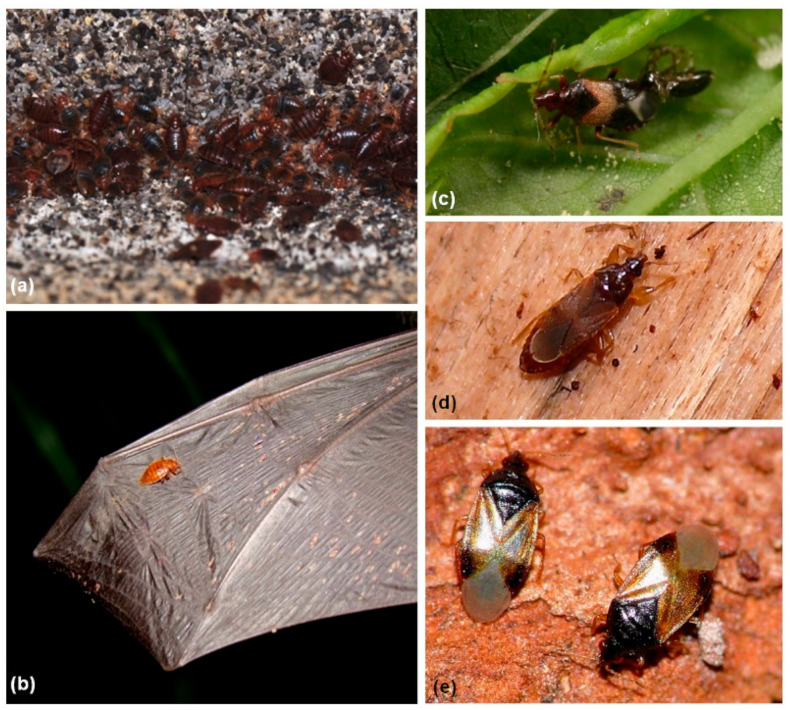
Habitus photographs of cimicoid species. (**a**) *Cimex pipistrelli* aggregating in a roost in Točník Castle (Bohemia, Czech Republic) where *Myotis myotis* (greater mouse-eared bat, Vespertilionidae) dwells; (**b**) *C. pipistrelli* feeding on *Nyctalus noctula*’s wing (common noctule, Vespertilionidae) (Slovakia; photo taken by M. Celuch); (**c**) just-emerged *Anthocoris chibi* on *Artemisia princeps* var. *orientalis* (Korea); (**d**) *Lasiochilus japonicus* dwelling under the bark of a dead oak tree (Korea); (**e**) *Orius minutus* during winter hibernation under the bark of *Zelkova serrata* (Ulmaceae) (Korea).

**Figure 2 insects-14-00267-f002:**
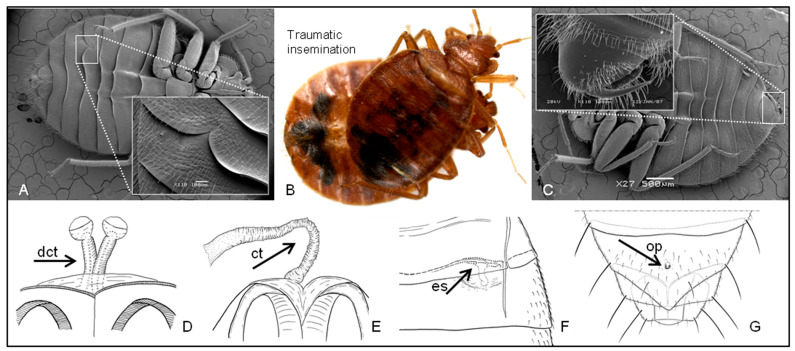
A copulation scene of traumatic insemination (TI) and various types of paragenitalia (PG) and male paramere. (**A**, **C**) Scanning electron microscope of *Cimex lectularius*; ectospermalege and male paramere are magnified in dotted box. (**B**) TI between male (upper) and female (lower) (*Cimex lectularius*). (**D**) *Blaptostethus aurivillus*, representing double copulatory tube (dct) (modified after Yamada (2008) [17]); (**E**) *Scoloposcelis albodecussata*, representing copulatory tube (ct) (modified after Yamada and Hirowatari (2005) [18]); (**F**) *Xylocoris cerealis* representing ectospermalege (es) (modified after Yamada et al. (2006) [19]); (**G**) *Buchananiella pericarti* representing omphalus (op) (modified after Yamada and Yasunaga (2009) [20]).

**Figure 3 insects-14-00267-f003:**
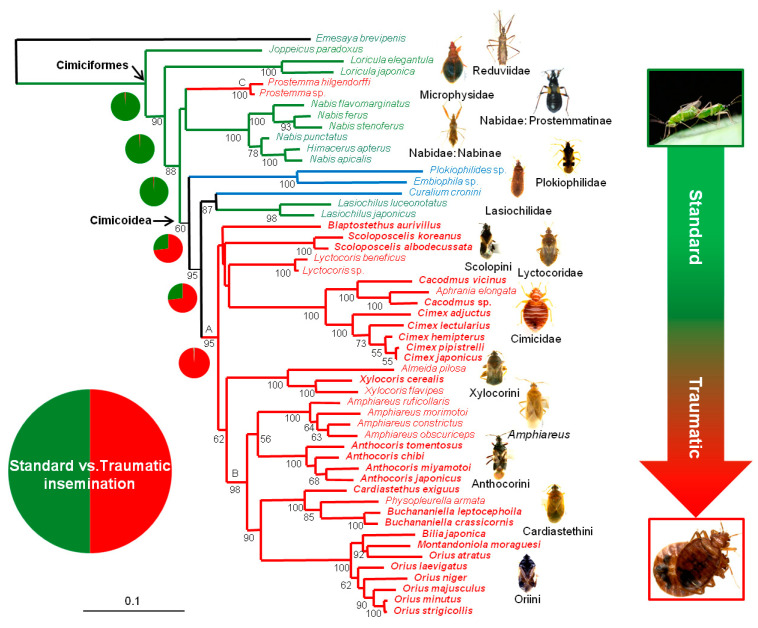
Maximum likelihood tree and ML bootstrap support values (>50%). Traumatic insemination (TI) clades and taxa highlighted in red; standard insemination clade and taxa highlighted in green. Uncertain taxa and clades in terms of their copulating habits highlighted in light blue. Taxa in bold indicate whose females evolved PG. Reconstructions of ancestral copulating method using BayesTraits are indicated as pie charts that shows the posterior probabilities of each type at the respective nodes representing higher-taxa relationships.

**Table 1 insects-14-00267-t001:** GenBank accession numbers and character coding of copulation habits and paragenitalia types.

Family	Subfamily/Tribe	Species	Copulation Type	Paragenitalia Type	Collecting Country	Accession Number			
						18S rRNA	28S rRNA	16S rRNA	COI
Ingroup									
Anthocoridae	Oriini	*Orius atratus*	TI	CT	Japan	GQ258414	GQ258449	GQ258387	GQ292177
		*Orius laevigatus*	TI	CT	The Netherlands	GQ258416	GQ258451	GQ258371	GQ292148
		*Orius niger*	TI	CT	Nepal	GQ258418	GQ258453	GQ258392	GQ292182
		*Orius majusculus*	TI	CT	USA	JQ782789	JQ782811	JQ782758	JQ782832
		*Orius minutus*	TI	CT	Republic of Korea	GQ258417	GQ258452	GQ258372	GQ292157
		*Orius strigicollis*	TI	CT	Republic of Korea	GQ258420	GQ258455	GQ258374	GQ292146
		*Bilia japonica*	TI	CT	Nepal	GQ258406	GQ258439	GQ258363	JQ782816
		*Montandoniola moraguesi*	TI	CT	Republic of Korea	GQ258413	GQ258448	GQ258370	-
	Anthocorini	*Anthocoris chibi*	TI	CT	Republic of Korea	GQ258403	GQ258437	GQ258362	GQ292164
		*Anthocoris miyamotoi*	TI	CT	Republic of Korea	GQ258405	GQ258438	GQ258361	GQ292152
		*Anthocoris tomentosus*	TI	CT	USA	JQ782774	JQ782810	JQ782755	JQ782815
		*Anthocoris japonicus*	TI	CT	Republic of Korea	GQ258404	GQ258436	GQ258360	GQ292142
	Cardiastethini	*Amphiareus ruficollaris*	TI	None	Japan	GQ258394	GQ258430	GQ258383	GQ292169
		*Amphiareus obscuriceps*	TI	None	Republic of Korea	GQ258393	GQ258429	GQ258358	GQ292178
		*Amphiareus constrictus*	TI	None	Japan	GQ258397	GQ258427	GQ258359	GQ292170
		*Amphiareus morimotoi*	TI	None	Japan	GQ258398	GQ258428	GQ258361	GQ292174
		*Buchananiella crassicornis*	TI	OP	Malaysia	GQ258407	GQ258441	GQ258364	GQ292144
		*Buchananiella leptocephala*	TI	OP	Malaysia	GQ258408	GQ258442	GQ258365	JQ782817
		*Physopleurella armata*	TI	None	Republic of Korea	GQ258421	GQ258456	GQ258375	GQ292167
		*Cardiastethus exiguus*	TI	OP	Republic of Korea	GQ258409	GQ258443	GQ258366	GQ292165
	Scolopini	*Scoloposcelis albodecussata*	TI	CT	Japan	GQ258422	GQ258457	GQ258376	GQ292128
		*Scoloposcelis koreanus*	TI	CT	Republic of Korea	GQ258423	GQ258458	GQ258377	GQ292130
	Xylocorini	*Xylocoris cerealis*	TI	ES	Thailand	GQ258395	GQ258459	GQ258384	GQ292172
		*Xylocoris flavipes*	TI	None	Thailand	JQ782790	JQ782795	JQ782756	JQ782835
	Almeidini	*Almeida pilosa*	TI	None	Thailand	JQ782793	JQ782794	JQ782754	JQ782814
	Blaptostethini	*Blaptostethus aurivillus*	TI	DCT	Malaysia	GQ258400	GQ258440	JQ782772	-
Lyctocoridae		*Lyctocoris beneficus*	TI	None	Republic of Korea	GQ258412	GQ258447	GQ258369	JQ782826
		*Lyctocoris* sp.	TI	None	Cambodia	JQ782786	JQ782804	JQ782757	JQ782827
Lasiochilidae		*Lasiochilus japonicus*	SI	None	Republic of Korea	GQ258410	GQ258445	GQ258367	GQ292184
		*Lasiochilus luceonotatus*	SI	None	Japan	GQ258408	GQ258446	GQ258368	JQ782825
Cimicidae	Cimicinae	*Cimex lectularius*	TI	ES	USA	JQ782782	JQ782797	JQ782771	JQ782823
		*Cimex adjunctus*	TI	ES	USA	JQ782778	JQ782801	JQ782767	JQ782820
		*Cimex hemipterus*	TI	ES	Malaysia	JQ782779	JQ782802	JQ782768	JQ782821
		*Cimex pipistrelli*	TI	ES	Czech Republic	JQ782780	JQ782803	JQ782770	JQ782824
		*Cimex japonicus*	TI	ES	Japan	JQ782781	JQ782796	JQ782769	JQ782822
	Cacodminae	*Cacodmus vicinus*	TI	ES	Egypt	JQ782777	JQ782800	JQ782766	JQ782819
		*Cacodmus* sp.	TI	ES	Morocco	JQ782776	JQ782799	JQ782765	JQ782818
		*Aphrania elongata*	TI	None	Algeria	JQ782775	JQ782798	JQ782764	-
Curaliidae		*Curalium cronini*	uncertain	uncertain	USA	EU683128	-	-	-
Plokiophilidae	Plokiophilinae	*Plokiophiloides* sp.	uncertain	None	Laos	JQ782792	JQ782813	-	-
	Plokiophilinae	*Embiophila* sp.	uncertain	None	Thailand	JQ782791	JQ782812	JQ782773	-
Outgroup									
Microphysidae		*Loricula pilosella*	SI	None	Republic of Korea	GU194610	GU194685	GU194532	GU194763
		*Loricula elegantula*	SI	None		EU683151	AY252577	EU683098	-
Nabidae	Nabinae	*Nabis stenoferus*	SI	None	Republic of Korea	GQ258426	GQ258434	GQ258379	GQ292211
		*Nabis apicallis*	SI	None	Cambodia	JQ782783	JQ782806	JQ782761	JQ782828
		*Nabis punctatus*	SI	None	Republic of Korea	JQ782785	JQ782807	JQ782763	JQ782830
		*Nabis ferus*	SI	None	Cambodia	JQ782784	JQ782805	JQ782762	JQ782829
		*Nabis flavomarginatus*	SI	None	Republic of Korea	GQ258424	GQ258433	GQ258380	GQ292213
		*Himacerus apterus*	SI	None	Republic of Korea	GQ258425	GQ258435	GQ258381	GQ292205
	Prostemmatinae	*Prostemma* sp.	TI	None	Cambodia	JQ782788	JQ782809	JQ782760	JQ782834
		*Prostemma hilgendorfii*	TI	None	Cambodia	JQ782787	JQ782808	JQ782759	JQ782833
Joppeicidae		*Joppeicus paradoxus*	SI	None	USA	EU683147	EU683200	EU683094	AY252951
Reduviidae		*Emesaya brevipennis*	SI	None	USA	EU683139	AY252560	AY252796	EU683231

Numbers beginning with “JQ” were directly sequenced by this study; other numbers are from Jung et al. (2010) [3]; Jung, Duwal, and Lee (2011) [29]. Type coding mainly follows references and direct observations (see Materials and Methods). Abbreviations: TI, traumatic insemination; SI, standard insemination; ES, ectospermalege; CT, copulatory tube; DCT, double copulatory tube; OP, omphalus.

**Table 2 insects-14-00267-t002:** Primer sets used in this study.

Region	Primer	Sequence	Annealing Temp.	Reference
COI	LCO1490 HCO2198	GGTCAACAAATCATAAAGATATTGG TAAACTTCAGGGTGACAAAAAATCA	43.5–48 °C	[32]
18S rRNA	18S-1 18S-4 18S-2 18S-3	CTGGTTGATCCTGCCAGTAGT GATCCTTCTGCAGGTTCACC AGATACCGCCCTAGTTCTAACC GGTTAGAACTAGGGCGGTATCT	48–50 °C	[33]
28S rRNA	28S-DD 28S-FF	GGGACCCGTCTTGAAACAC TTACACACTCCTTAGCGGAT	45–50 °C	[34]
16S rRNA	16S-A 16S-B	CGCCTGTTTAACAAAAACAT CCGGTTGAACTCAGATCA	45–50 °C	[35,36]

## Data Availability

All the data generated in this work were provided in the article.

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
