# Peer review of "Molecular Phylogeny of Cimicoidea (Heteroptera: Cimicomorpha) Revisited: Increased Taxon Sampling Reveals Evolution of Traumatic Insemination and Paragenitalia"

_insects, 2023, doi:10.3390/insects14030267_

Round 1

Reviewer 1 Report

Dear Editor,

This study added some new molecular data of more taxa of Cimicoidea, although not very much, and analysis the phylogenetic relationships of families and some tribes of Anthocoridae. They confirmed some monophyletic groups or clades proposed in the previous studies, although this study doesn't well resolve the phylogenetic relationship of Cimicoidea, may be due to the less molecular data, even some nodal supports are weak in BI and ML analyses, especially the tribe relationships of Anthocoridae in BI analyses are not well resolved. The authors also explored the evolutionary history of traumatic insemination and paragenitalia in Cimicoidea.

In general, the authors added more taxa and more data, got the evolutionary route of traumatic insemination and paragenitalia, it is still  worth to publish it.

Some questions as follows:

1)    Why is it very different between the results of ML and BI analyses?

2)    Line 23. Please write the full name of TI When it first appeared.

3)    Line 131. Please summarize the number of new sequences of each molecular fragment.

4)    Line 162-175. The primers can be listed in a table. Supply brief description about the primers in the text.

5)    Line 191. “gene was” changes to “genes were”.

6)    Line 143. The legend of Table 1 is CT and DCT, but all CP and DCP are appeared in Table 1.

7)    Table1, Embiophilinae has already been downgraded to subtribe Embiophilina, and was placed into tribe Plokiophilini by Schuh RT, Štys P, Cassis G, Lehnert M, Swanson D, Bruce TF. (2015) New genera and species of Plokiophilidae from Australia, Fiji, and Southeast Asia, with a revised classification of the family (Insecta: Heteroptera: Cimicoidea). American Museum Novitates, 3825, 1–24.

Author Response

Response to reviewer 1,

This study added some new molecular data of more taxa of Cimicoidea, although not very much, and analysis the phylogenetic relationships of families and some tribes of Anthocoridae. They confirmed some monophyletic groups or clades proposed in the previous studies, although this study doesn't well resolve the phylogenetic relationship of Cimicoidea, may be due to the less molecular data, even some nodal supports are weak in BI and ML analyses, especially the tribe relationships of Anthocoridae in BI analyses are not well resolved.

- Response to reviewer 1:

Yes, we agreed. Thank you for the valuable review and feedback. 

The authors also explored the evolutionary history of traumatic insemination and paragenitalia in Cimicoidea.

- Response to reviewer 1:

Yes, it is the one of our results. Thank you.   

In general, the authors added more taxa and more data, got the evolutionary route of traumatic insemination and paragenitalia, it is still worth to publish it.

- Response to reviewer 1:

Thank you for your evaluation.

Some questions as follows:

1)    Why is it very different between the results of ML and BI analyses?

- Response to reviewer 1:

This difference may result from the models applied to each analysis. We addressed this issue by adding the sentence on the reason of difference. Please check the correction in 4.1 in Discussion section. Thank you.

2)    Line 23. Please write the full name of TI When it first appeared.

- Response to reviewer 1:

Yes, we agreed. We corrected this as you pointed out. Thank you.   

3)    Line 131. Please summarize the number of new sequences of each molecular fragment.

- Response to reviewer 1:

Yes, we agreed. We corrected this by adding the summary of sequences of each gene in 2.1 in Materials and Methods. Please check the corrections. Thank you.   

4)    Line 162-175. The primers can be listed in a table. Supply brief description about the primers in the text.

- Response to reviewer 1:

Yes, we agreed. We addressed this issue by adding the table 2 for the primer information. Thank you.

5)    Line 191. “gene was” changes to “genes were”.

- Response to reviewer 1:

Yes, we agreed. It is corrected. 

6)    Line 143. The legend of Table 1 is CT and DCT, but all CP and DCP are appeared in Table 1.

- Response to reviewer 1:

CT and DCT are correct. We corrected them as you pointed out. Thank you.

7)    Table1, Embiophilinae has already been downgraded to subtribe Embiophilina, and was placed into tribe Plokiophilini by Schuh RT, Štys P, Cassis G, Lehnert M, Swanson D, Bruce TF. (2015) New genera and species of Plokiophilidae from Australia, Fiji, and Southeast Asia, with a revised classification of the family (Insecta: Heteroptera: Cimicoidea). American Museum Novitates, 3825, 1–24

- Response to reviewer 1:

Yes, we agreed. We corrected this by changing Embiophilinae into Plokiophilinae. Thank you.   

Reviewer 2 Report

Dear authors MS seems be in line but few points below are need to take in consideration before to run into the 2nd version

1. state in Table 1. if any of these species were previously used in past contributions (might be indicated by an asterisk) to support such increasing in taxa sampling. Or if none of these are included here then add it as another suppl material

2. primers used can be summarized in an extra suppl table, no need to be in line with text.

3. figure 2, increase  quality of image, its flurry

4. all 3 topologies must be in full discussed with previous analyses made, I strongly encourage enriching discussion in point 4.2 by comparing these results to old ones to increase audiences and get more straightforward the poiot to do this study

Author Response

Response to reviewer 2,

MS seems be in line but few points below are need to take in consideration before to run into the 2nd version

- Response to reviewer 2:

Yes, we agreed. Thank you for your valuable review and excellent feedback. 

  1. state in Table 1. if any of these species were previously used in past contributions (might be indicated by an asterisk) to support such increasing in taxa sampling. Or if none of these are included here then add it as another suppl material

- Response to reviewer 2:

In order to indicate the new generated sequences or the sequences used in past contributions, we used the prefix of the accession number, and the relevant sentence was already shown under the table 1. Therefore, we did not correct this. Please check this information.

  1. primers used can be summarized in an extra suppl table, no need to be in line with text.

- Response to reviewer 2:

Yes, we agreed. We corrected this by adding the table 2 for primer information. Please check this in the revised script. Thank you.   

  1. figure 2, increase  quality of image, its flurry

- Response to reviewer 2:

All figures were made in a high quality and were submitted; however, the quality of figures in MS looks not that high. We will address this issue by providing the original high quality figures in the next process. Thank you.  

  1. all 3 topologies must be in full discussed with previous analyses made, I strongly encourage enriching discussion in point 4.2 by comparing these results to old ones to increase audiences and get more straightforward the poiot to do this study

- Response to reviewer 2:

We already indicated the discussions on the incongruence between the results of previous study (old) and the present study in the model-based analyses and in parsimony analysis. However, as you suggested, we addressed this by indicating the additional discussion between the analyses in the previous study and the results in the present study. Please check the correction in 4.2. Thank you.

Reviewer 3 Report

This manuscript is in good form, well written and presented. It’s important paper based on a huge and representative material, but unfortunately, according to Table 1 most of them are results of other researchers (Jung et al., 2010; Jung, Duwal & Lee, 2011). In my opinion, more figures (photos and drawings) on taxons observed by authors can improve the level of quality of this manuscript. I suggest one necessary correction – redrawing Figure 3. It is illegible, the letters are too small, and the taxon names aren’t readable.

Overall, the paper is a valuable item in the scientific context and contains relevant scientific information. I warmly recommend publishing it in your journal.

Author Response

Response to reviewer 3,

This manuscript is in good form, well written and presented.

- Response to reviewer 3:

Yes, we agreed. Thank you for the valuable evaluation and feedback. 

It’s important paper based on a huge and representative material, but unfortunately, according to Table 1 most of them are results of other researchers (Jung et al., 2010; Jung, Duwal & Lee, 2011).

- Response to reviewer 3:

The materials (with prefix GQ) in other previous contributions were also provided by the same author (Jung S) in this research. Therefore, we did not correct this issue. Thank you.  

In my opinion, more figures (photos and drawings) on taxons observed by authors can improve the level of quality of this manuscript.

- Response to reviewer 3:

We selected the representatives of the morphological and biological characteristics for this research, and instead, we chose to cite the references on the relevant information, not showing too many characters, due to the page limitation. Therefore, we did not address this issue. Thank you.  

I suggest one necessary correction – redrawing Figure 3. It is illegible, the letters are too small, and the taxon names aren’t readable.

- Response to reviewer 3:

Figure 3 was made in a high quality and are enough expandable to read the small letters (supporting values, and scientific names); however, the quality of figures in MS for review looks not that high. We will address this issue by providing the original high quality figures in the next process. Thank you.  

Overall, the paper is a valuable item in the scientific context and contains relevant scientific information. I warmly recommend publishing it in your journal.

- Response to reviewer 3:

Yes, we agreed. Thank you so much for the valuable review and feedback.